# On Effect of Chloroform on Electrical Activity of Proteinoids

**DOI:** 10.3390/biomimetics9070380

**Published:** 2024-06-23

**Authors:** Panagiotis Mougkogiannis, Andrew Adamatzky

**Affiliations:** Unconventional Computing Laboratory, UWE, Bristol BS16 1QY, UK; andrew.adamatzky@uwe.ac.uk

**Keywords:** thermal proteins, proteinoids, microspheres, unconventional computing

## Abstract

Proteinoids, or thermal proteins, produce hollow microspheres in aqueous solutions. Ensembles of the microspheres produce endogenous spikes of electrical activity, similar to that of neurons. To make the first step toward the evaluation of the mechanisms of such electrical behaviour, we decided to expose proteinoids to chloroform. We found that while chloroform does not inhibit the electrical oscillations of proteinoids, it causes substantial changes in the patterns of electrical activity. Namely, incremental chloroform exposure strongly affects proteinoid microsphere electrical activity across multiple metrics. As chloroform levels rise, the spike potential drops from 0.9 mV under control conditions to 0.1 mV at 25 mg/mL. This progressive spike potential decrease suggests chloroform suppresses proteinoid electrical activity. The time between spikes, the interspike period, follows a similar pattern. Minimal chloroform exposure does not change the average interspike period, while higher exposures do. It drops from 23.2 min under control experiments to 3.8 min at 25 mg/mL chloroform, indicating increased frequency of the electrical activity. These findings might lead to a deeper understanding of the electrical activity of proteinoids and their potential application in the domain of bioelectronics.

## 1. Introduction

Thermal polymerisation of mixtures of amino acids result in the formation of dynamic oligopeptide structures—proteinoids—that tend to grow continuously [1,2,3]. These structures also assemble into larger aggregates at both microscopic and macroscopic scales [4,5,6,7,8]. Proteinoids, found within these structures, display an intricate emergent chemistry such as imitating ion channels, regulating voltage, and responding to stimuli in a manner similar to living cells [9,10,11,12]. Proteinoids are bioinspired materials that can mimic the biophysical and self-organisational characteristics observed in living organisms [1,13,14,15,16,17].

While previous research focused on morphological characteristics [2,18,19,20], recent advances in electrochemical analysis provide critical insights into bioelectric activity [21]. Evidence suggests that proteinoid aggregates have significant self-propagating electrical currents with dynamic oscillations resembling neuronal potentials [22,23,24,25].

Proteinoids exhibit a specific susceptibility to chemical stimuli that causes alterations in their morphology [26,27,28]. Changes in both the structural and electrostatic binding affect the tertiary conformations and likely the properties of ion permeability [29,30].

The proteinoid microsphere produces spikes remarkably similar to the action potential generated by neurons [23,24] (Figure 1). The precise mechanisms underlying proteinoid excitability are not universally agreed upon. The physical conditions governing excitability are delineated in [31]. One theoretical model posits that excitability is contingent on the strength of coupling between basic and acidic proteinoids, which undergo abrupt changes over time due to material flow [8]. Additional models worth considering include the classical model involving ionic gradients established through channels [32] and the generation of membrane potential through the immobilisation of mobile ions. In this model, ions adhere to the surface of proteinoid microspheres, thereby generating a nonzero membrane potential [33]. To evaluate if the excitability of the proteinoids relates to the excitability of living substrates, we decided to conduct experiments with chloroform. Another rationale for conductive experiments discussed presently is to evaluate how close proteinoid microspheres are to living forms. Here we follow Claud Bernard’s hypothesis that the sensitivity of anaesthetics differentiates living substrates from inanimate chemical forms [34].

Chloroform is a chemical compound employed as an anaesthetic, which effectively inhibits nerve signals and induces paralysis in certain parts of the mammals’ body [35,36]. Chloroform acts as an anaesthetic not only on mammals and their single cells [37] but also on plants [38,39,40,41], fungi [42], slime moulds [43], and bacteria [44].

Chloroform is believed to exert its effects by permeating cell membranes, according to researchers [45,46]. These membranes possess channels that facilitate the passage of ions and messages. When chloroform becomes trapped in the membranes, it obstructs the channels, impeding the normal transmission of signals [47,48].

Through the examination of chloroform’s impact on synthetic cell membranes, we can enhance our understanding of these phenomena [49,50]. Membranes were synthesised utilising “proteinoids”, which are polypeptides composed of amino acids. The proteinoids have characteristics similar to cell membranes, forming small vesicles [51].

Efficient signal transmission is vital for the optimal functioning of nerve cells. Therefore, it is crucial to understand techniques for restoring conduction in order to maintain optimal health. In addition to its medical applications, studying membrane signal plasticity and adaptive mechanisms can also contribute to unconventional computing [52,53]. Exploring proteinoid excitability phenomena [23,54] could unlock exciting possibilities for innovative data-processing methods that utilise the intricate ionic processes [55,56] found in living organisms. More precisely, manipulating the restore/collapse spiking in tailored biopolymer systems could offer a pathway for conducting multistate logic operations. Phase transitions that are influenced by the environment can potentially be used to convert chemical signals into digital data, affecting the rhythm and appearance of the signals. Designing robust and adaptive bioelectronic systems involves cataloguing emergent identity resiliency factors. These systems are capable of self-recovering vital communication pathways, even in the face of disruption threats. Efforts to rescue delicate biological conduction phenomena have led to foundational understandings that ignite bioinspired innovation in sensing, computing, and tissue engineering [57,58]. By looking at signal stability from a different perspective, this research gains even more promise and urgency. Figure 2 presents a schematic overview of the biomimetic system based on proteinoids and the effect of chloroform on their electrical properties. The mind map illustrates the main components of the system, including arginine, glutamic acid, and the formation of dipeptides. It also highlights the key properties of proteinoids, such as their catalytic and enzymatic capabilities, amphiphilic nature, and self-assembly into microspheres.

In this study, we explore the effects of varying chloroform concentrations on the dynamic electrochemical indicators in thermal proteinoid polymers. We measure shifts in oscillatory rhythms, periodicity, spike intensity, and duration across a range of exposures, from trace amounts to anaesthetic concentrations. The insights gained provide clarity on tuning ranges for sustained, regulated bioelectronic phenotypes in the presence of solvent disruption, thereby guiding the development of tunable signal-processing materials [59].

Proteinoids, comprised of thermal amino acid polymers, have been shown to self-assemble into cell-like microspheres able to exhibit life-like phenomena. Yet, an enhanced understanding is still needed of the underlying structural dynamics that enable proteinoids’ biomimetic activities. Advanced computational modelling, as shown for a small L–glutamic acid:L–arginine system, can help reveal mechanisms related to spontaneous peptide aggregation, organisation, and conformational responses in proteinoids by simulating their molecular behaviour.

## 2. Methods and Materials

The amino acids L-Aspartic acid and L-Glutamic acid were procured from Sigma-Aldrich, St. Louis, MO, USA ensuring a purity level exceeding 98%. Upon procurement, the thermal polycondensation technique put forth by Mougkogiannis et al. [4] was implemented for the synthesis of proteinoids. This procedure comprised heating equimolar mixtures of the amino acids at 180 °C throughout a nitrogen atmosphere. The heating process was carried out for 30 min with consistent stirring. Postheating, the proteinoids were subjected to lyophilisation and stored at an ambient temperature. For the experiments undertaken, chloroform was procured from Sigma-Aldrich. It is listed under the CAS Number 67-66-3, possessing a molecular weight of 119.3 g/mol. An FEI Quanta 650 microscope was used to capture the morphological characteristics of the proteinoids via scanning electron microscopy (SEM). Notably, the gold sputter-coating procedure that is commonly employed in SEM imaging to enhance conductivity was skipped, underscoring the inherent conductive properties of proteinoids even under high-vacuum conditions. I–V characteristics curves were recorded with the employment of a Keithley 2450 sourcemeter. Platinum–iridium-coated stainless steel subdermal needle electrodes, procured from Spes Medica S.r.l. (Genova, Italy), were implanted into the proteinoids, allowing a nearly 10 mm distance between each electrode pair. Moreover, the electrodes’ electrical activities were chronicled accurately by the use of a high-resolution data logger strapped with a 24-bit analog-to-digital converter ADC-24 from Pico Technology, St. Neots, UK. The chronoamperometry measurements integral to our study were carried out using an Anapot EIS potentiostat.

Figure 3 illustrates the setup used for measuring the electrochemical properties of the proteinoid–chloroform samples. The system employs needle electrodes made from platinum- and iridium-coated stainless steel wires, which are placed 10 mm apart in the proteinoid sample. A high-precision 24-bit ADC data logger is used to capture the voltage responses, while a synchronised heating block allows for the simultaneous monitoring of thermal and electrical parameters. The setup enables the detection of small voltage fluctuations in the μV range, providing insights into the potential dynamics of proteinoids under various conditions.

## 3. Results

Polymerised microspheres were subjected to step-wise chloroform exposure regimes, and the resulting bioelectric responses were recorded to assess the impact of chloroform on proteinoid activity dynamics. The intrinsic spiking potentials of control proteinoid assemblies reflected baseline excitation characteristics. The addition of chloroform-soaked filter paper gradually increased the vapour-phase concentrations in the container. To generate an escalating exposure series, 0.5 × 0.5 cm, 1 × 1 cm, and 3 × 3 cm filter sheets were saturated with chloroform and suspended over proteinoid solutions. Finally, direct chloroform addition into proteinoid samples at 25 mg/L investigated responsiveness to solvated phases.

This range encompasses anaesthetic concentrations for aquatic organisms, allowing for the examination of neurological excitation effects. Being denser than water, the chloroform likely partitions into lower layers, thereby inducing gradient diffusion. The systematic escalation in chloroform levels allows for the quantification of concentration-dependent modulatory impacts on proteinoid interspike intervals and spike morphologies. Contrasting control dynamics with responses across exposure regimes provides insights into intrinsic and externally triggered influences on emergent bioelectric behaviours in this thermal protein polymer system.

Scanning electron microscopy elucidated both the baseline ultrastructure (and the impact of solvent exposure on proteinoid morphologies, as exemplified in Figure 4a–c). However, when exposed to chloroform, the broken microspheres developed bursting membranes and hollow interiors (Figure 4d–f). This microscopy evidence supports the hypothesised swelling and permeation effects.

### 3.1. Computational Analysis: Molecular Mechanics Simulation of Chloroform and Dipeptide Systems

Molecular mechanics energy minimisation was conducted to determine the optimal energy configuration of the interaction between chloroform and the dipeptide L-Glu-L-Arg. The energy-minimisation process involves optimising the molecular geometry by reducing the overall potential energy of the system. In molecular mechanics, the potential energy function is commonly represented as the summation of different energy components, including bond stretching, angle bending, torsional angles, and nonbonded interactions. The total potential energy ($E_{\mathrm{total}}$) can be represented as
(1)$$E_{\mathrm{total}}=E_{\mathrm{stretch}}+E_{\mathrm{bend}}+E_{\mathrm{torsion}}+E_{\mathrm{non}–\mathrm{bonded}}$$
where $E_{\mathrm{stretch}}$ is the energy associated with bond stretching, $E_{\mathrm{bend}}$ is the energy due to angle bending, $E_{\mathrm{torsion}}$ represents the torsional energy, and $E_{\mathrm{non}–\mathrm{bonded}}$ accounts for the nonbonded interactions, such as van der Waals and electrostatic interactions. The energy-minimisation algorithm uses an iterative process to precisely adjust the atomic coordinates in order to minimise the overall potential energy of the system. The minimisation process persists until convergence requirements are met, such as achieving a minimum gradient norm or reaching the maximum number of iterations. The energy-reduction process in this study concluded at iteration 2511 due to the gradient norm reaching a value lower than the minimal gradient norm. The energy-minimisation results offer valuable insights into the stability and favourable interactions between chloroform and the dipeptide L-Glu-L-Arg. An analysis was conducted to determine the contributions of different energy components to the overall energy of the system in its minimum configuration. The results are presented in Table 1. The simulation findings provide an in-depth analysis of the energy landscape and the variables that affect the interaction between the dipeptide and chloroform during the energy-minimisation approach. Figure 5 illustrates the molecular representations and energy minimisation of the interaction between chloroform and the dipeptide L-Glu-L-Arg. The initial position of chloroform (${\mathrm{CHCl}}_{3}$) and the dipeptide L-Glu-L-Arg before energy minimisation is depicted in Figure 5a. The minimum energy configuration obtained after the molecular mechanics calculations (iteration 2511) is illustrated in Figure 5b. The gradient norm was less than the minimum gradient norm, indicating that the energy minimisation converged normally. The system’s total energy at the minimum configuration is −10.4245 kcal/mol, which is a result of the contributions of a variety of energy components, including stretch, bend, stretch–bend, torsion, non-1,4 van der Waals interactions, 1,4 van der Waals interactions, and dipole–dipole interactions (Table 1). Figure 6 illustrates the outcomes of the molecular mechanics simulation for the interaction between chloroform and the dipeptide L-Glu-L-Arg. Figure 6a illustrates the progression of the minimisation algorithm and the root mean square (RMS) gradient in relation to the iteration number. Figure 6b illustrates the relationship between the iteration number and the RMS move, which represents the atomic displacements. The final energy values at iteration 2511 (Figure 6c) illustrate the system’s numerous energy components’ cumulative contributions to the total energy. The simulation results offer insights into the energy landscape and the stability of the interaction between the dipeptide and chloroform during the energy-minimisation process.

### 3.2. Baseline Electrical Dynamics

According to the spike analysis, the oscillations recurred with an average periodicity of 1392.86 s (Figure A1 and Figure 7a) with amplitudes ranging from 0.36 mV to 1.91 mV (μ = 0.89 mV; σ = 0.42 mV), Figure A2d.

The amplitude intensity distribution was positively skewed (skewness = 1.37) and leptokurtic (kurtosis = 3.91), indicating a lengthy right tail of high amplitude spikes that deviated from the mean. This shows that the dynamics of proteinoid excitation entail occasional significant bursts mixed with more constant suppressed oscillations.

Control proteinoid electrical activity exhibits complexity and stochasticity, as seen in the signal shape variations in Figure A1 across time and across repeated measurements. Profiling spike shape attributes could further provide insight into how microscale structural assembly events appear in the reported macroscale potential patterns.

Under control conditions in the absence of chloroform, the spiking activity of L–Glu:L–Arg proteinoid samples varied significantly (Figure A2a). The interspike intervals measured varied from 595.24 to 1952.38 s, with a mean of 1392.86 ± 106.46 s (Figure A2a). The distribution has a negative-skew Gaussian-like shape (skewness = −0.36) and a low kurtosis (1.95). The heterogeneous rhythm suggests that proteinoid oscillations develop from complex dynamics even when no external perturbations are present. Changes in these distributions in response to alterations such as the addition of chloroform may reveal environmental influences of proteinoid bioelectric behaviour. Identifying characteristics that modulate rhythmic spiking could provide insight into how to programme and improve the interface with these synthetic mimics of excitable membranes.

Figure A5 compares the potential dynamics of a typical proteinoid with that of water. The pattern depicted in Figure A5a illustrates the notable difference in the electrical activity of the proteinoid and water. The proteinoid displays prominent and well-defined peaks in electrical potential over a period of time, whereas water demonstrates a rather steady response with negligible variations. In order to highlight the minimal increase in the spiking activity of water, a magnified section of the graph is shown in Figure A5b. This detailed observation verifies that water generates insignificant spikes, which are reduced in both intensity and frequency as compared to the proteinoid. The results highlight the distinct electrical characteristics of the proteinoid system and its ability to produce and transmit electrical signals, which are not present in water under similar conditions.

### 3.3. Vapour-Phase Chloroform Exposures

#### 3.3.1. Modulation of Proteinoid Spiking upon Exposure to 0.5 cm × 0.5 cm Chloroform-Embedded Filters

Our analysis of Figure 7b and Figure A2b reveals intriguing findings regarding the response of proteinoids to the incorporation of chloroform in the vapour phase. We observe a reduction in spike amplitudes and a shift in interpulse intervals, indicating that the protein system regulates internal excitation dynamics when faced with external perturbation.

Notably, the distribution of amplitudes shows a decrease in variability while maintaining a positive skewness. This suggests that the conductivity deviations are limited, yet the overall activity remains outward. Additionally, we observe shortened periods and accelerated rhythms, indicating the generation of actionable potentials to counter the disruptive effects of diffusive chemicals. In essence, we find that the proteinoid architectures exhibit self-stabilisation reflexes when confronted with low-level environmental interruptions.

The baseline mean spike potential of control proteinoid samples (Figure A2) was 0.89 ± 0.42 mV. The addition of a 0.5 × 0.5 ${\mathrm{cm}}^{2}$ chloroform source caused observable alterations in spiking activity. When exposed to filter-sourced vapours, the average spike potential was 0.281 ± 0.016 mV. This is a 68% reduction in spike intensity when compared to the control spike intensity without chloroform. The significant potential attenuation supports feasible chloroform diffusion gradients reaching proximal proteinoid microspheres and incorporating them into assemblies.

The continuous pattern of spiking activity, on the other hand, indicates preserved bioelectric functioning. The further incremental scaling of vapour-phase chloroform via enlarged filter sheets or new sources may intensify adaptive dynamics dose-dependently. Comparing morphological and conductivity readouts across various exposure scopes may help to identify mechanisms that allow resonant oscillations to continue despite solvent interference. The early spike potential alterations show significant proteinoid spiking flexibility that is accessible to external electrochemical manipulation.

#### 3.3.2. Modulation of Proteinoid Spiking upon Exposure to 1.0 cm × 1.0 cm Chloroform-Embedded Filters

Exposure to chloroform for an extended period of time triggers distinct patterns of proteinoid responses, as illustrated in Figure 7c and Figure A3b. The intervals contract by more than 98%, decreasing from an average of approximately 1300 s to a range of 20–30 s (Figure 7c). This hyper-excitable regime suggests crossing critical thresholds leads to increased conductivity and uncontrollable signal waves. Investigating the exact timing and vapour concentrations that induce hyper-excitability can reveal insights into the development of sensor materials with rapid-burst capabilities. Meanwhile, the confined spike amplitude ranges show a consistent decrease with increased chloroform insulation, measuring 0.104 ± 0.011 mV upon 1 × 1 ${\mathrm{cm}}^{2}$ exposure (Figure A3b). The distribution appears to be skewed, indicating a consistent level of proteinoid excitability despite limited intensities. The combination of acute temporal sensitisation and regulated amplitude adaptation highlights the diverse proteinoid signal plasticity. By delving deeper into the textural changes and binding interactions that lead to functional modifications, we can gain a better understanding of the principles that govern vapour-induced fluctuation dynamics. Understanding the crucial tuning parameters can allow for deliberate activation of specific proteinoid states, tailored for personalised bioelectronic applications.

The significant decrease in interspike intervals after prolonged exposure to chloroform in the surrounding environment is measured and presented in Figure A3b. The proteinoid spiking period was measured to be 923.00 ± 107.62 s, indicating a significant decline of over 30% compared to the previous localised exposure. The distributions display characteristics of a left-skewed Gaussian distribution with reduced kurtosis. The sudden change in spiking indicates a shift towards a highly stimulated state, with the rate of spikes increasing by more than five times. As indicated by the period metrics (Figure A3b), the maximum intervals decrease to 1360 s, while the minimums approach 230 s between spikes. Understanding the timeframe and levels of vapour concentration that trigger this heightened activity will shed light on the underlying mechanisms of this complex phenomenon. The proteinoid systems display a remarkable sensitivity to the accumulation of chloroform over time. By delving deeper into the morphologies of resultant spikes and analysing gating responses, valuable insights can be gained regarding the deliberate induction of rapid-burst exocycles, which can be highly advantageous for specific biosensing applications.

#### 3.3.3. Modulation of Proteinoid Spiking upon Exposure to 3.0 cm × 3.0 cm Chloroform-Embedded Filters

Increasing the concentration of chloroform in the air on 3 × 3 ${\mathrm{cm}}^{2}$ filter paper causes continuing changes in proteinoid spiking properties. Positive outliers in potential (Figure 7d and Figure A3a) may reflect these changes while preserving regular oscillatory patterns (Figure A2d). Spike potentials show a significant decrease, with average values of 0.091 ± 0.008 mV, suggesting a reduction of nearly 90% when compared to the control. The potential distribution shifted to the right indicates a significant change in conductivity; however, there are brief exaggerations that exceed the maximum values of the control. Under these conditions, the time intervals between spikes likewise decrease (Figure A2d), with average values of around 1000 s. The fact that there is a continuous pattern even at the lowest duration of approximately 550 s and the longest duration of approximately 1500 s, on the other hand, illustrates the durable nature of these artificial stimulation systems in the presence of vapour-phase disturbances. Analysing the morphological changes in waveform adaptations observed by statistical profiling would provide a better understanding of the tuning principles that govern long-term stimulus responsiveness.

### 3.4. Solvated Chloroform Response Dynamics

Exposing thermal proteinoid polymers directly to chloroform at a concentration of 25 mg/L caused significant but controlled changes in the patterns of emerging excitation waves (Figure 7 and Figure A3). The average spike amplitude detected 0.065 ± 0.009 mV, demonstrating 90% attenuation relative to baseline (Figure A3d). Simultaneously, the frequency became highly unpredictable with occasional contraction, reaching temporary intervals as short as 69.80 s (Figure 7). Despite the disturbance, there are consistent and rhythmic fluctuations that demonstrate the impressive ability of proteinoids to recover and adapt (Figure 7e). These fluctuations are presumably a result of self-adaptation mechanisms that actively maintain conductive structures. Elucidating the precise order of binding interactions, structural changes, and controlled shifts in conductance that contribute to the observed functional persistence may enhance the development of adaptable bioelectronics materials based on biological principles. Gradient titrations can also uncover key concentration thresholds that trigger either uncontrolled collapse or preserved excitability. Utilizing limited but viable ranges creates possibilities for deliberate and adjustable destabilisation, which in turn activates quick and intense sensory capabilities that are helpful in identifying specific organic substances. Ultimately, the demonstrated adaptive plasticity of proteinoids provides fundamental templates for developing computer circuits that are artificially robust and responsive to the environment.

### 3.5. Concentration-Dependent Effects

Table 2 and Table 3 summarise important measured proteinoid spiking potentials and periodicity properties throughout steadily increasing chloroform exposures encompassing the vapour phase and direct solvation introductions.

As demonstrated in Table 2, the average spike intensity gradually decreases with an increasing chloroform concentration, with mean values decreasing from 0.895 mV to 0.065 mV from control to 25 mg/mL conditions, indicating almost 90% damping.

Simultaneously, amplitude distribution widths narrow significantly, restricted to smaller ranges between peak and minimal spikes. Similarly, increased chloroform levels cause a shortening of interspike intervals (Table 3), reflecting increased excitation states. In essence, spike compression in conjunction with rhythmic coordination highlights a suppression of bioelectric disorder in the face of increasing solvent perturbation.

When comparing the control and 25 mg/L groups, the periodicity kinetics increase by more than 80%. Sharper rhythmic coordination emerges in conjunction with compressed waveform volatility, indicating more systematic ion conductance underneath the waveforms. Nonetheless, sustained spiking demonstrates persisting proteinoid bioelectricity with non-Gaussian statistical features, indicating retained nonlinear dynamics despite dose escalation up to solvent saturation. Following morphological changes at chloroform junctions can now reveal particular mechanisms that support regulated waveform adaptation.

As shown in Figure 8, boxplots summarise the distribution of key spiking metrics across incrementally elevated chloroform regimes. For interspike periods, control proteinoids exhibit the broadest spread spanning 595.24 to 1952.38 s. This likely reflects intrinsic bioelectric variability in the absence of external disruption. Upon initial localised vapour exposure (0.5 ${\mathrm{cm}}^{2}$), marginal interval tightening appears, progressively contracting further under more extensive coverings (1 ${\mathrm{cm}}^{2}$ and 3 ${\mathrm{cm}}^{2}$). Finally, direct solvent contact intensely heightens rhythmic rates, driving mean periods down to 228.2 ± 223.91 s.

Similarly for amplitudes, higher-intensity distributions prevail in control samples up to 0.9 mV, sequentially declining under rising chloroform presence down to 0.065 ± 0.028 mV. Tighter, lower-intensity profiles emerge denoting constrained excitation and pathway conduction.

Figure 8 thus quantifies chloroform’s dose-dependent impacts on concentrating emergent rhythms yet attenuating signal output, pointing to reoptimisation not collapse. Further interrogation of the compensatory mechanisms sustaining modulated waveforms could direct the fabrication of resilient bioelectronic materials.

#### 3.5.1. Tuning Resistance and Reactance in Proteinoid Assemblies through Chloroform Incorporation

In Figure 9, sweeping proteinoid sample exposure to varied input signal frequencies reveals key electrical traits altered upon chloroform incorporation. Most prominently, a logarithmic decline in dynamic capacitance manifests in both control and exposed samples, with mean values of 0.05 nF and 0.087 nF, respectively. The heightened variability and range compression with chloroform suggest regulated rather than disrupted energy storage modalities. Similarly, impedance spectra progress logarithmically from 0.115 ± 0.875 Ω in the control state down to 1.3956 × $10^{-3}$ ± 6.4324 × $10^{-4}$
Ω when exposed. In effect, the programmed proteinoid systems exhibit the adaptive self-correction of electrical characters rather than waveform collapse when confronted with chloroform disturbances. Further elucidation of the precise binding mechanisms and morphological adaptations eliciting enduring, regulated bioelectric identities could direct the fabrication of resilient, specialised conductive materials.

#### 3.5.2. Concentration-Dependent Modulation of Proteinoids–Chloroform Memristive Dynamics Characterised by Chronoamperometry

Chronoamperometry reveals the brief electrical responses of proteinoid systems, as illustrated in Figure 10 for L–glutamic acid:L–arginine under an increasing chloroform concentration. The memristive spike and recovery dynamics show that solvent interactions cause transitory conductivity enhancement.

Figure 10 depicts the time-dependent current for various situations, with time on the *x*-axis and current on the *y*-axis. The present responses indicate various patterns, indicating the effect of proteinoid exposure to chloroform under varied settings.

Let us analyse the experimental results in the framework of imitated neural networks. Figure 11 shows an analysis of the neural networks’ initial synaptic weights. The heat maps of generated synaptic weight matrices operate as a symbolic representation of the patterns of neural connections that could arise in unconventional computing systems based on proteinoids. These 2D visualisations illustrate potential variations in neuronal organisation under different chemical conditions by representing proteinoid suspensions as a concrete neuromorphic substrate, albeit the notion is still in its early stages. For example, the noticeable variations in colour intensity and gradient distributions may indicate the formation of different network structures and collective behaviours. We expect to observe more clustering in control situations, while chloroform exposure is likely to result in higher randomness. Continuing simulations that incorporate these patterns of connection and function into models of spiking neural networks can assist in forecasting the emerging computing abilities. Furthermore, by establishing a correlation between the characteristics of the heat map and the observed conductivity behaviours and recovery durations in experiments (as shown in Figure 11), we can obtain insights into the relationship between short-term and long-term plasticity manifestations and the development of simulated architecture. In addition to static visualisation, the inclusion of temporal changes towards stable end states can enhance our understanding of how proteinoid systems maintain a balance between order, chaos, and adaptable bioderived computing.

Equation (Equation 2) can express the current transition as follows:(2)$$I\left(t\right)=\frac{dQ}{dt}$$
where I(t) signifies the current and *Q* symbolises the charge. One can calculate the current by differentiating the charge with regard to time. The shifting current patterns under varied settings (shown in Figure 10) indicate a change in the charge flow rate due to variable levels of chloroform exposure.

The synaptic weight matrix (*W*) can be first expressed in neural network modelling as
(3)$$W_{ij}=2*rand\left(\right)-1$$
where ($W_{ij}$) represents the synaptic weight from the (i)th presynaptic neuron to the (j)th postsynaptic neuron and rand() returns a random value between −1 and 1. As illustrated in Figure 11, the effect of chloroform exposure causes a clear difference in the initialisation of synaptic weights.

The simulation of a neural network was implemented by generating artificial neurons and connecting them with “synaptic” weights. For each condition analysed, we created a network including 30 simulated neurons. The neural activity was correlated with binary temporal coding, which was obtained from the time-dependent current signals seen in our investigations. In order to create these temporal codes, a sampling threshold of $1\;\times \;10^{-4}$
μA was established. Any recorded current signal above this threshold was considered as neural activity and designated as a state of ‘1’. Subsequently, we randomly chose a subset of these data points, making sure that the size of the group was inversely related to the number of artificial neurons. This approach added a realistic aspect to our model by incorporating the intrinsic unpredictability found in biological systems. We assigned a set of initial synaptic weights to each neuron in the network. These weights were determined arbitrarily as random values within a range of −1 to 1. The weights were specifically intended to adapt themselves iteratively according to a predetermined learning rule, allowing the network to learn progressively over time. The initial allocation of these synaptic weights was shown as a heatmap for each condition, providing us with a visual understanding of the starting state of the network prior to any learning taking place.

Finally, the chloroform appears to have a distinct influence on the electrical properties of the proteinoid, which is detectable through changes in the current and seen in the variation of synaptic weights in our model’s neural network.

Neuronal currents exhibit varied distribution characteristics under the five different situations, as shown in Figure 12. Notably, the mean current increases incrementally from the control condition to the 3 ${\mathrm{cm}}^{2}$ chloroform exposure condition. However, when proteinoids are exposed to chloroform at a concentration of 25 mg/L, the mean current drops significantly to around 3046 μA. The boxplot’s quartiles and whiskers provide a detailed view of the data dispersion and probable outliers for each scenario. The median neuronal current has a similar pattern to the mean current. The significant decrease in the median current upon chloroform exposure demonstrates the significant impact of chloroform on the electrical characteristics of the proteinoid.

The Lempel–Ziv (LZ) complexity is a measure that quantifies the level of randomness in finite sequences or patterns [61]. As observed in the synaptic weight maps (Figure 11), the LZ complexity declined as the chloroform exposures increased up to 3 ${\mathrm{cm}}^{2}$, and then showed a minor recovery at 25 mg/L. The control instance demonstrated the highest LZ score of 4.7399, suggesting a pattern that is almost maximally complex. A decrease in the LZ complexity indicates an increase in the organisation and predictability of the created connection graphs. This indicates that when solvent concentrations increase, the architecture of neurons tends to shift from disorder to a more organised and localised clustering. Nevertheless, once a specific limit is surpassed, the excessive disruption of proteinoids can lead to the degradation of organised assembly. In general, the fluctuations in nonmonotonic LZ complexity are associated with the possibility of disturbance followed by the restoration of self-organised circuit development when influenced by chloroform. The quantitative complexity metrics enhance the analysis of induced folding modifications and their organisational propagation throughout architectural development, providing better insights than visual map inspection alone. Table 4 summarises the Lempel–Ziv complexity results.

The schematic (Figure 13) is designed to visually represent the overall trend of increased order that emerges in the self-organised dynamics of proteinoids as the chloroform levels increase. The purpose behind generating the binary tree abstraction, as depicted in Figure 13, lies in its ability to cognitively embody the nonlinear association between the morphological complexity of adjustable proteinoids and the observed concentration levels of chloroform exposure during experimentation. To be specific, an initially careful and controlled addition of a minuscule amount of the solvent contributes significantly to the formation of a highly consistent and orderly structure throughout the process of molecular self-assembly. This is reflected in a marked decrease in the branching complexity. Notwithstanding, this process of simplification begins to retrogressively reverse at higher saturation levels, suggesting that supersaturation adversely impacts the organised dynamics of the system. We thereby put forth the proposition that proteinoid systems exhibit a labyrinthine configuration optimisation landscape, influenced by interactions that do not necessarily adhere to a linear pattern. Furthermore, the visualisation, while qualitative, inspires quantitative investigations into precisely defined solution parameters for optimally hatching desired bioarchitectures—whether dendrimers, fractals, or other uniform meshes ideal for unconventional computing.

Nevertheless, the particular intricacy of branching patterns represents one way to quantify the randomness of patterns, but there are other valid interpretations of the model as well. We believe that the importance of this lies in demonstrating the possibility of using chemical parameter alterations alone to guide proteinoid structural bifurcations towards desired computational geometries. By enhancing solution control and gaining a deeper understanding of how macroscale assembly is influenced by molecular triggers, it is possible to create proteinoid architectures that closely resemble basic living cells.

### 3.6. Elucidating Input–Output Relationships of Chloroform-Tuned Proteinoid Network Biointerfaces under Periodic Voltage Harmonic Forcing

Complex signals and harmonic analysis are tools for quantifying nonlinear system behaviours. Any periodic waveform can be represented as a sum of distinct harmonic frequency components:(4)$$x\left(t\right)=\sum_{k=1}^{\infty }A_{k}\sin\left(k\omega t+\phi_{k}\right)$$
where $A_{k}$ and $\phi_{k}$ are the harmonic amplitude and phase, respectively; ω is the base frequency; and *k* indexes each harmonic component.

Figure 14 illustrates that solvent-modulated proteinoid systems produce a diverse range of harmonic outputs when exposed to different voltage waveforms. The emerging higher-order harmonics exhibit nonlinear transformational reactions that are similar to neural circuits [62,63]. Tracing the source of particular harmonics allows us to gain an understanding of the corresponding molecular processes occurring in their natural environment—such as ion movement, membrane polarisation, conformational changes, or structural rearrangements. In the end, utilising input harmonics that efficiently drive certain proteinoid harmonic outputs could potentially allow for precise computational filtering and pathway gating in bioderived unconventional computing.

The rich mosaic of harmonic content embedded within the observed proteinoid response signals, as depicted in Figure 14, shows that similarly elaborate processing is possible. Isolating the origins of unique harmonic signatures remains a challenge. However, the diversity implies that protonic, electronic, and conformational movements all interact and contribute in different ways. Excitingly, using multifrequency harmonic forcing as a computational tuning parameter may soon allow for the manipulation of desirable collective dynamics.

## 4. Potential Molecular Mechanisms

Chloroform has been extensively used as an inhalation anaesthetic. It causes organisms to lose consciousness by entering and affecting the lipid bilayer membranes, thereby disrupting important signalling pathways [64]. Studies have specifically demonstrated that chloroform interferes with specialised lipid raft microdomains that are crucial for the localisation of enzyme activity and cell communication mechanisms [65,66,67]. The indiscriminate buildup of chloroform in membranes hinders the proper functioning of enzymes such as phospholipase D2 (PLD2), which regulates brain signalling molecules. Moreover, the focusing of substances at the membranes of nerve cells disrupts ion channels such as TREK–1, which necessitate accurate control in order to transmit electrical signals that are fundamental to cognitive function and alertness. By utilising these established methods of membrane disruption, chloroform and similar anaesthetic chemicals can effectively reduce consciousness and brain activity that is crucial for awareness and cognitive functions in living organisms.

Positioning it in stark contrast to living cell membranes, where chloroform exposure results in the disruption of ion channels and the eventual electrical isolation of the bilayer, the proteinoid microspheres tested within this study were observed to maintain voltage oscillations, despite being subjected to similar levels of solvent exposure. This striking incongruity suggests that there exist fundamental differences between the mechanisms of excitation of the two involved systems. The persisting harmonics displayed by the proteinoids hint at the dominance of bulk effects arising from various factors—the pressure strains brought upon by swelling, the formation of temporary pores, and alterations in the dynamics of molecular clustering—over the inhibition of ion movement. In a proteinoid system, the small soluble chloroform molecules may penetrate the peptide matrix, which has been thermally cross-linked, in their attempts to fragment it. This action results in the oscillations in the proto-cell wall, causing it to alternately contract and expand.

How can chloroform cause proteinoid microspheres to conduct and oscillate, unlike lipid bilayer cell membranes, which are insulating?

As a preliminary attempt to generate a hypothesis, we offer potential explanations that focus on the permeation of the proteinoid matrix, the phenomena associated with swelling, the disruption of interior dynamics, along with the absence of surface ion channel analogues.

Potential mechanisms are:Transient pore formation.Small chloroform molecules infiltrate and cause temporary defects in cross-linked peptide matrix.Swelling strain effects.Proteinoid microsphere wall dimensions increase or decrease with solvent attack.Modulated molecular clustering.Disrupts interior packing dynamics and charge mobility pathways.Lack of ion channel disruption.Unlike cells, no insulation from the impedance of selective ion flow.Bulk matrix effects rather than surface phenomena.Deep permeation into peptide structure underpins observed dynamics.

This in-depth understanding of the modus operandi of proteinoid microspheres can be vitally instrumental in deciphering their versatility and potential utilities in the realm of unconventional computing. It propels the investigation a step forward towards the efficient utilisation and manipulation of these bioarchitectures in the development and designing of innovative computational systems.

## 5. Discussion

The results of our research provide compelling evidence that increasing chloroform levels progressively alter the electrical signalling of proteinoids. We observed the occurrence of two distinct patterns.

First, an alteration was made to the dimensions of the spikes. Increasing the amount of chloroform results in a progressive reduction in the size of the spikes, with a decrease of almost 90% compared to normal levels observed at the highest tested concentrations. This can be considered as an equivalent to going to sleep, as happens with living substrates.

Second, the interspike intervals decrease as the dosage of chloroform increases.

These findings indicate that proteinoid systems exhibit an adaptive response to chloroform exposure rather than undergoing complete degradation. In the field of biology, adaptations indicate an ability to withstand and recover from disruption. The proteinoids exhibit recovery against disruption up to a certain threshold. Their chemical identity remains stable, as evidenced by the modulation of spike characteristics. This might be seen as the analogue of the remarkable ability of living cells to rapidly regenerate structures in response to damage.

By understanding the precise mechanisms by which proteinoids sustain electrical functionality despite damage, we can unveil novel approaches to engineer highly resilient computers by drawing inspiration from distinctive biological concepts. More precisely, we may utilise the same mechanisms that enable proteinoids to adjust their signals in response to chloroform in order to construct systems that can endure disturbance. By emulating the mechanisms via which proteinoids regulate signal intensity, synchronise time, and redirect channels as needed, it might be possible to design computers that ensure the uninterrupted transmission of information, even under severe attack.

There was no recorded direct chemical reaction between chloroform ($\begin{array}{c} {\mathrm{CHCL}}_{3} \end{array}$) and the dipeptide L–Glu:L–Arg. However, in a biological context, chloroform is a well-known membrane disruptive agent, and it is normally not reactive with amino acid side chains under physiological conditions. However, contact can occur indirectly when the solvent disrupts the folding and secondary structure of proteinoids, including L–Glu:L–Arg. The dominant interaction pathway is through noncovalent, solvent-mediated perturbation, rather than direct covalent modification:(5)$$\mathrm{Glu}–\mathrm{Arg}+n{\mathrm{CHCl}}_{3}\overset{∆H_{\mathrm{solute}}}{\to }\mathrm{Glu}–{\mathrm{Arg}}_{(\mathrm{CHCl}3)n}$$
where the binding of *n* chloroform molecules to L–Glu:L–Arg proteinoid aggregates alters the enthalpic state ($\Delta H_{\mathrm{solute}}$) via proteinoid solvation and conformational change.

Rather than reactive chemistry, the impacted secondary structure and folding disrupts control state dynamics:(6)$$\mathrm{Glu}–{\mathrm{Arg}}_{(\mathrm{CHCl}3)n}\overset{\mathrm{config}\cdot \mathrm{change}}{\to }\mathrm{altered}–\mathrm{Glu}–\mathrm{Arg}+n{\mathrm{CHCl}}_{3}$$

With first-order rate dynamics restoring native baseline structures over subsecond timescales after chloroform evacuation.

Figure 15 depicts the influence of chloroform exposure on the electrical characteristics of proteinoid microspheres. The image shows two states: the control proteinoids with prominent spikes (+) before anaesthesia on the left, and the subdued spheres after chloroform treatment on the right. The chloroform molecules, depicted as triangles, are superimposed over the proteinoid population that was anaesthetised, visually illustrating the inhibitory relationship. The arrows indicate that the electrical muting effect is a result of exposure to chloroform. The text box outlines the significant effects of anaesthesia that were deduced from experimental findings, including the cessation of neural activity, prevention of external disturbances, restoration of normal function once the anaesthesia wears off, and preservation of the physical structure. This visualisation enhances the stimulus–response data by highlighting the effectiveness of carefully adjusted chloroform in managing previously difficult-to-handle spiking proteinoids for purposes of characterisation and patterning.

The comparative analysis in Table 5 shows numerous major contrasts in functional capacities between pure and anaesthetised proteinoids relevant to their use in unconventional computation applications. Pure proteinoids exhibit inherent electrical signalling and associated communication channels, which are disrupted by anaesthesia, leading to a disturbance in both the spontaneous activity and network connectivity. This compromise significantly enhances their ease of manipulation, since anaesthesia allows for direct handling, placement, and integration without being hindered by the need for insulation to protect against signal interference effects. Significantly, the effects of anaesthesia can be reversed, as proteinoids restore their electrical spiking properties once it is removed. The halted spiking behaviours are further supported by the quantitative spike train statistics provided. These recorded variations demonstrate how anaesthesia effectively limits the information capacity of proteinoids, making them more versatile in applications that require flexibility, such as biosensor design and the development of patterned circuits. Additional research can enhance the optimisation of anaesthetic exposure in order to maintain enough processing for certain use scenarios.

Finally, although anaesthesia allows for the manipulation of proteinoids, the exact mechanism by which it disrupts electrical activity is not well understood. The proposed elements of the chloroform–proteinoid mechanism path are as follows:The direct binding of chloroform molecules within hydrophobic pockets on proteinoid surfaces, structurally distorting excitable domains.Changes in the ion permeability and transport kinetics of the membrane, altering the spike propensity.Conformational shifts of intrinsic pore-forming subunits, temporarily blocking conductive states.The insertion of chloroform into lipid bilayers, modifying membrane dielectric properties and capacitance.Combinatorial effects on proteinoid morphology and physiology, disrupting synchronised oscillations.

Finally, chloroform, which has been previously used as an anaesthetic, could offer substantial potential in the field of biochemistry and unconventional computing. The agent’s ability to disrupt membranes can modify the folding and secondary structure of proteins, such as proteinoids. This deviation from normal protein behaviour has the potential to be utilised in unconventional computing to influence the processing and storage of information. Proteinoids, which are synthetic proteins that have similar traits to natural proteins, are highly potential contenders for biological computation [68,69]. Exposure to chloroform causes notable alterations in their electrical characteristics, potentially influencing the development and operation of biocomputational devices. The incorporation of the biocomputational modelling of proteinoids interacting with chloroform might be regarded as a distinctive element of biological computing, wherein biological systems regulate or carry out computational processes. Moreover, this specific biochemical interaction, which results in changes in neuromorphic signals and synthesis processes, might potentially be investigated as a subfield within the realm of biological computing, particularly in the context of encoding data.

It is noted that the spikes recorded in proteinoids may seem small compared to the spiking activity commonly associated with neurons. However, our approach is more similar to EEG measurements, which capture the overall electrical activity of the system. The observed patterns and dynamics are highly significant and offer crucial insights into the electrical characteristics of proteinoids. Moreover, an important finding in our research is the impact of chloroform on the spiking patterns of proteinoids. We demonstrated that the presence of chloroform has a substantial impact on electrical activity, resulting in noticeable variations in both the frequency and amplitude of the spikes. This observation indicates that proteinoids are sensitive to chemical perturbations and have the potential to be used as a model system for investigating the interplay between chemical agents and electrical signalling in biological systems. The results emphasise the unique electrical properties of the proteinoid system and its capacity to generate and convey electrical impulses, which are absent in water under similar conditions. Our research establishes the fundamental basis for future investigations into the electrical characteristics of proteinoids and their potential use in areas such as biosensors and unconventional computing.

From a chemical perspective, we can expand the methodology in this research to investigate other dipeptide-derived systems and their interactions with various organic solvents. It is important to understand how dipeptides and solvents interact at the molecular level in order to explain how the electrical activity and shape of proteinoid structures change. For more research, it is important to use advanced computer methods like molecular dynamics simulations and quantum chemical calculations to fully understand the forces between molecules, changes in conformation, and energy landscapes connected to these interactions. Moreover, the present study emphasises the necessity of thoroughly examining and describing the physical and chemical features of proteinoid–solvent systems. Different experiments, like spectroscopy (FTIR and Raman), calorimetry, and small-angle X-ray scattering (SAXS), can tell us a lot about how solvents change the structure and function of dipeptides. By integrating experimental results with theoretical modelling, we can achieve a deep understanding of the structure–property interactions in proteinoid-based systems. The results of this study also affect prebiotic chemistry and the role of organic chemicals in life’s origin. The experiment illustrating chloroform’s effect on proteinoids’ electrical behaviour indicates that the interaction with organic compounds may have had significant effects on the development of basic functional systems. For more information on how different chemicals affect the features and functions of proteinoid structures, more research could help us understand how primitive life forms might have evolved. The aims and objectives of this paper were to investigate the macro behaviour of proteinoid ensembles affected by anaesthesia with chloroform. A deeper understanding of the intrinsic mechanisms of interactions between dipeptides and chloroform could be a key direction for future research. The characterisation will include magnetic resonance microscopy and the use of Gaussian, AMBER, and CHARMM chemical modelling packages for the precise modelling of molecular interactions, predicting properties such as binding energy, preferred conformations, and interaction energies.

## 6. Conclusions

By conducting experiments on programmable bioabiotic substrates, researchers can effectively lead the development of new hardware architectures that have adaptive characteristics. Specifically, by reproducing the observed processes involved in amplitude modulation, time recalibration, and pathway remapping, we can provide a foundation for specialised neuromorphic computers that can maintain stable performance even in the face of external disturbances or attempts to disrupt information. This study uncovers uncharted opportunities within the realm of unconventional computing through an in-depth exploration of the correlation between instability and inventiveness in model biophysical systems. The quantification of spiking potential characteristics shows a progressive decrease in excitation heights and variability in response to sequential chloroform insertion, indicating a dose-dependent modulation of the proteinoid assembly network’s dynamic range. This supports the idea of a balance in these assemblies, as the initial robustness is gradually replaced by deterioration at amplified exposure levels like 25 mg/L, suggesting an optimisation pathway based on a balance between disorder induction and order reconstruction. The considerable reduction in the intervals between spikes suggests an increased processing speed, maybe due to chloroform infiltration-induced swelling instabilities. The network’s resilience in retaining coherency despite external perturbations reveals that proteinoid systems may adapt to changes and may promote new organisational patterns. Our findings can help direct engineering pursuits towards the development of hybridised biological–synthetic processors, uniquely designed for customised adaptation processes.

## Figures and Tables

**Figure 1 biomimetics-09-00380-f001:**
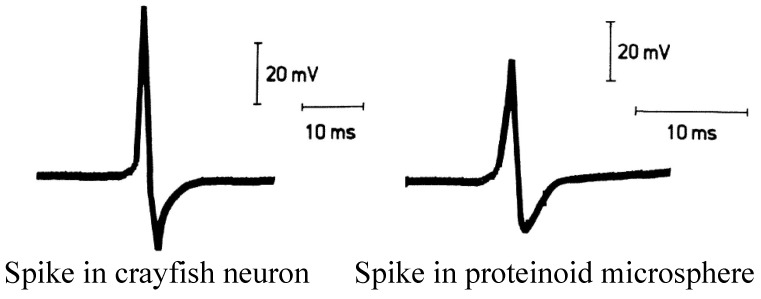
Action potentials from crayfish stretch-receptor neuron and from proteinoid microsphere. Modified from [22].

**Figure 2 biomimetics-09-00380-f002:**
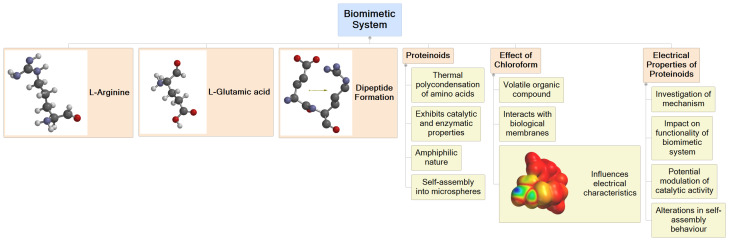
Schematic representation of the biomimetic system based on proteinoids and the effect of chloroform on their electrical properties. The main components of the system include arginine and glutamic acid, which undergo dipeptide formation. Proteinoids, formed through the thermal condensation of amino acids, exhibit catalytic and enzymatic properties, an amphiphilic nature, and the ability to self-assemble into microspheres. The introduction of chloroform, a volatile organic compound, into the system leads to interactions with the proteinoid membranes, influencing their electrical characteristics. The study focuses on investigating the mechanism behind these interactions and their impact on the functionality of the biomimetic system. Specifically, the effect of chloroform on the electrical properties of proteinoids is explored, including potential modulation of catalytic activity and alterations in self-assembly behaviour. Understanding the interplay between chloroform and proteinoids is crucial for elucidating the dynamics and adaptability of the biomimetic system in response to external factors. The insights gained from this study can contribute to the development of advanced functional materials and the understanding of the behaviour of proteins in biological systems.

**Figure 3 biomimetics-09-00380-f003:**
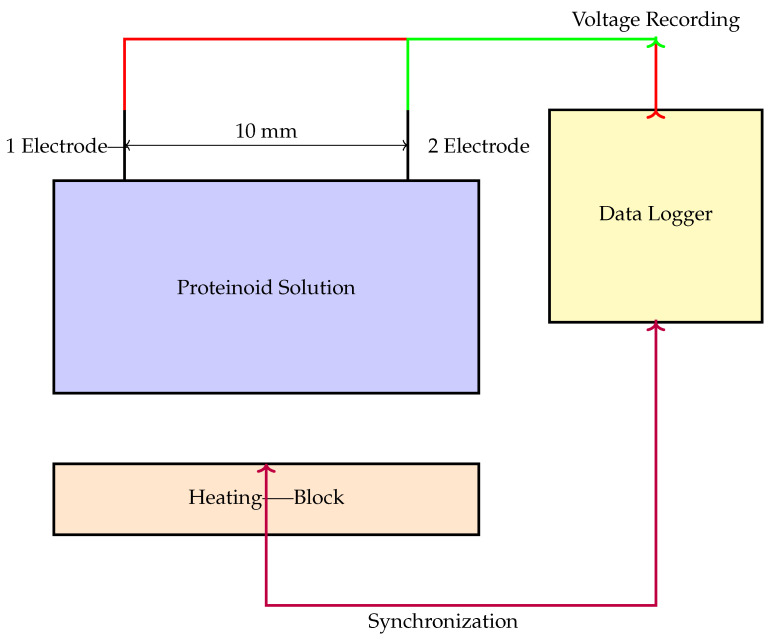
Schematic diagram of the experimental setup for electrochemical measurements of proteinoid samples. The proteinoid solution is placed in a container, and two needle electrodes (Electrode 1 and Electrode 2) made from platinum- and iridium-coated stainless steel wires are immersed in the solution, positioned 10 mm apart. The electrodes are connected to a high-precision 24-bit ADC data logger (Pico Technology) to record the voltage responses. The container with the proteinoid solution is placed on a heating block to control and monitor the temperature during the experiment. The heating block and the data logger are synchronised to simultaneously monitor thermal and electrical parameters. This setup enables the detection of small voltage fluctuations in the μV range, allowing for the mapping of spatiotemporal voltage responses in the proteinoid system.

**Figure 4 biomimetics-09-00380-f004:**
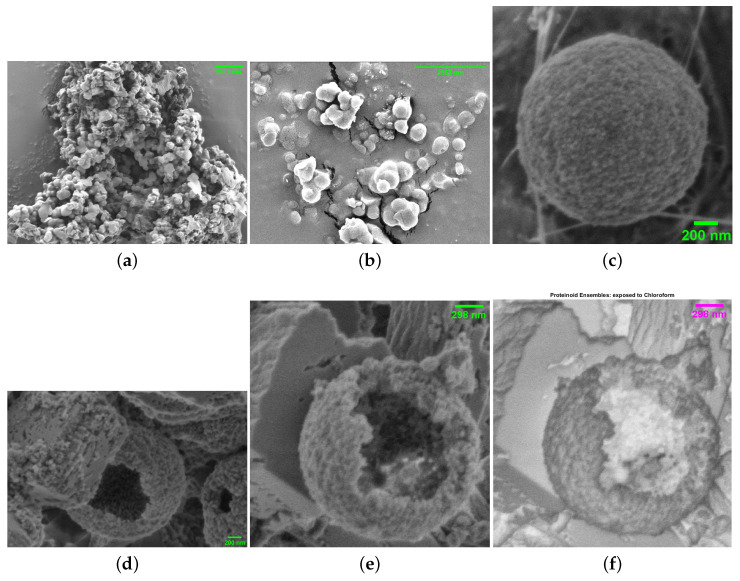
Scanning electron microscopy (SEM) images of proteinoid structures under different conditions. (**a**) Ensembles of proteinoids (scale bar: 2662 nm) exhibiting budding of nanospheres, indicative of reproduction. (**b**) Proteinoid nanospheres (scale bar: 2662 nm) with less evident reproduction compared to (**a**). (**c**) An isolated microsphere (scale bar: 200 nm). The samples in (**a**–**c**) were not exposed to chloroform. (**d**) Hollow microspheres (scale bar: 298 nm) exposed to chloroform. (**e**) The same microspheres as in (**d**) with a scale bar of 298 nm exposed to chloroform. (**f**) SEM image of the same microsphere as in (**e**) with inverted colours, highlighting the effects of chloroform exposure. The SEM images demonstrate the morphological changes in proteinoid structures, particularly the formation of hollow microspheres, upon exposure to chloroform. The budding of nanospheres in (**a**) suggests the potential for reproduction in proteinoid ensembles, while the isolated microsphere in (**c**) allows for detailed examination of individual structures. The comparison between unexposed and chloroform-exposed samples reveals the impact of chloroform on the morphology and internal structure of proteinoid microspheres.

**Figure 5 biomimetics-09-00380-f005:**
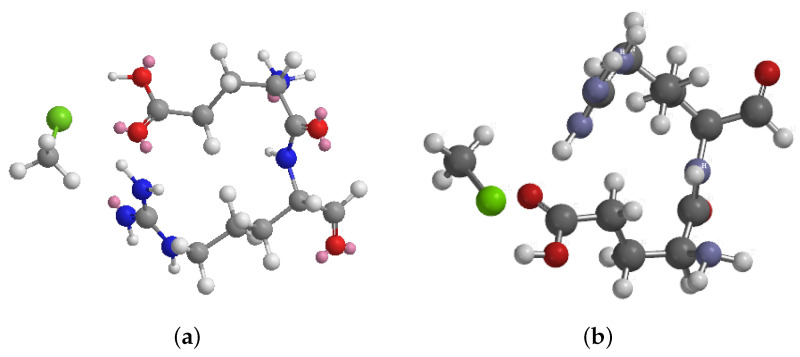
Molecular representations and energy minimisation of the interaction between chloroform and the dipeptide L-Glu-L-Arg. (**a**) Initial position of chloroform (${\mathrm{CHCl}}_{3}$) and the dipeptide L-Glu-L-Arg before energy minimisation. (**b**) Minimum energy configuration obtained after molecular mechanics calculations (iteration 2511). The energy minimisation converged normally, as indicated by the gradient norm being less than the minimum gradient norm. The total energy of the system at the minimum configuration is −10.4245 kcal/mol, with contributions from various energy components such as stretch (0.6971 kcal/mol), bend (5.7351 kcal/mol), stretch–bend (0.2346 kcal/mol), torsion (−0.5590 kcal/mol), non-1,4 van der Waals interactions (−9.7724 kcal/mol), 1,4 van der Waals interactions (7.9509 kcal/mol), and dipole–dipole interactions (−14.7107 kcal/mol). The energy-minimisation results provide insights into the stability and favourable interactions between chloroform and the dipeptide L-Glu-L-Arg. The AVOGADRO programme was utilised to conduct molecular dynamics simulations [60].

**Figure 6 biomimetics-09-00380-f006:**
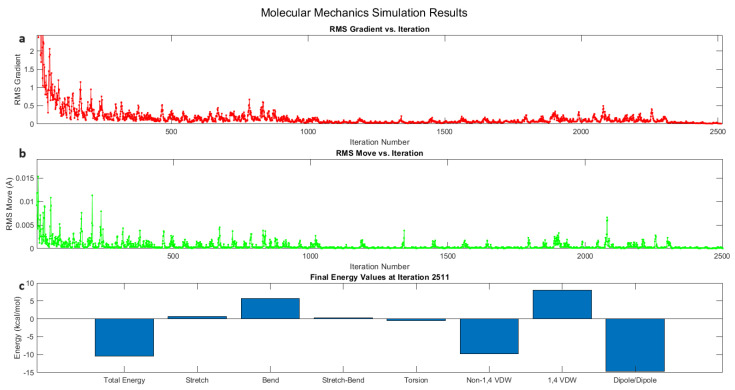
The results of the molecular mechanics simulation for the interaction between chloroform and the dipeptide L-Glu-L-Arg. (**a**) The progression of the minimisation algorithm and the root mean square (RMS) gradient as a function of the iteration number. (**b**) Plot of the iteration number against the RMS move, which denotes the atomic displacements. (**c**) The final energy values at iteration 2511, which illustrate the contributions of the system’s numerous energy components to the total energy. The primary contributors to the total energy at the minimum configuration are stretch (0.6971 kcal/mol), bend (5.7351 kcal/mol), stretch–bend (0.2346 kcal/mol), torsion (−0.5590 kcal/mol), non-1,4 van der Waals interactions (−9.7724 kcal/mol), 1,4 van der Waals interactions (7.9509 kcal/mol), and dipole–dipole interactions (−14.7107 kcal/mol). The simulation results offer a view of the energy landscape and the stability of the interaction between the dipeptide and chloroform during the energy-minimisation process.

**Figure 7 biomimetics-09-00380-f007:**
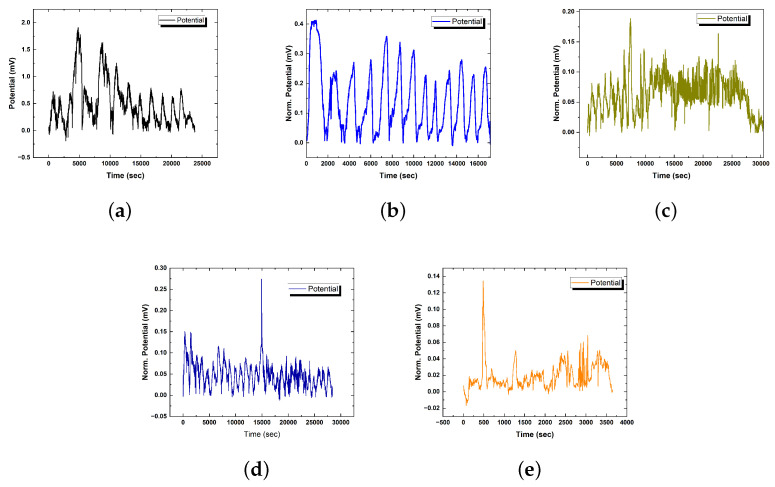
(**a**) The graph displays the temporal pattern of proteinoid spiking activity after applying a B–spline fit and removing the baseline signal. Through the extraction of slower field potential alterations, one can uncover and examine the dynamics of spike pattern. Spikes observed at around 5000 s and 7000 s have much higher densities in comparison to the intervals in between and subsequent to those timeframes. Modifications in spike characteristics such as intensity, breadth, or clustering due to external stimuli can provide insight into the adjustment of emergent excitation properties in these artificial polymers. This serves as the control sample showing baseline L–Glu:L–Arg proteinoid excitation dynamics without the effects of chloroform for comparison. (**b**) Proteinoid spiking amplitude dynamics with 0.5 × 0.5 ${\mathrm{cm}}^{2}$ chloroform filter paper exposure. When compared to greater baseline levels without chloroform, the timecourse indicates amplitude fluctuation over time (mean = 0.28 mV; max = 0.41 mV, min = 0.21 mV, σ = 0.06 mV). The distributions retain positive skew (0.87) but have lower kurtosis (2.83) compared to the control. Under limited chloroform, attenuation and confined amplitude ranges suggest dose-dependent regulation of excitation dynamics. (**c**) Examining the impact of extended chloroform (1 × 1 ${\mathrm{cm}}^{2}$ soaked filter paper) exposure on proteinoid spiking intervals. The graph depicts the fluctuations in bioelectric potential over a given period. At first, occasional exposure to chloroform vapour results in a reduction in the time interval between spikes, with an average of approximately 1300 s. Nevertheless, when exposed for an extended period of time surpassing 15,000 s, there is a notable decrease in intervals, shrinking to a mere 20–30 s, which signifies a contraction of more than 98%. This sudden change implies that the combined impact of chloroform might have exceeded specific thresholds, leading to increased conductance levels and hyper-excitable dynamics. Examining the precise time intervals and levels of exposure that trigger rapid bursting could yield valuable insights into the underlying mechanisms behind intentionally inducing high-frequency patterns, potentially leading to practical applications in sensing applications. (**d**) Bioelectric potential dynamics with increased exposure to a 3 × 3 ${\mathrm{cm}}^{2}$ chloroform-soaked filter. The plot shows a significant reduction in the mean spike amplitude of 0.091 ± 0.008 mV (minimum 0.049 mV; maximum 0.274 mV) across a 15 min period of modulated spiking oscillations in the plot. Positive skewness (2.99) and high kurtosis (12.93) in the data distribution point to an outlier-pronounced distribution at higher doses. (**e**) Spiking activity of proteinoids diluted with 25 mg/L chloroform. The recorded waveform displays changes in spiking patterns over a time period of 900 s, characterised by a notable decrease in mean amplitude of 0.065 ± 0.009 mV compared to control conditions. Variability in periodicity is also evident, with intervals ranging from 69.80 to 623.00 s. Interestingly, the observed oscillations persist despite the disruption, indicating the operation of inherent homeostatic mechanisms that maintain functional conduction pathways.

**Figure 8 biomimetics-09-00380-f008:**
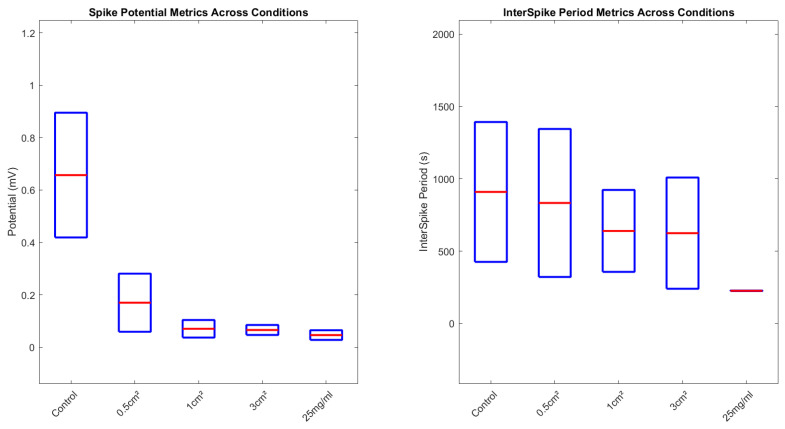
Interspike period measurements and spike potential metrics in proteinoids depicted across different chloroform exposure levels. Under control conditions and chloroform concentrations of 0.5 ${\mathrm{cm}}^{2}$, 1 ${\mathrm{cm}}^{2}$, 3 ${\mathrm{cm}}^{2}$, and 25 mg/mL, each boxplot represents the distribution of interspike intervals (in seconds) and the distribution of spike potentials (in millivolts). Each box delineates the IQR, with the middle line denoting the median value, demonstrating the data’s distribution and variability. The broad spread in the ‘Control’ and ‘1 ${\mathrm{cm}}^{2}$’ conditions signifies a wider variation in interspike period measurements.

**Figure 9 biomimetics-09-00380-f009:**
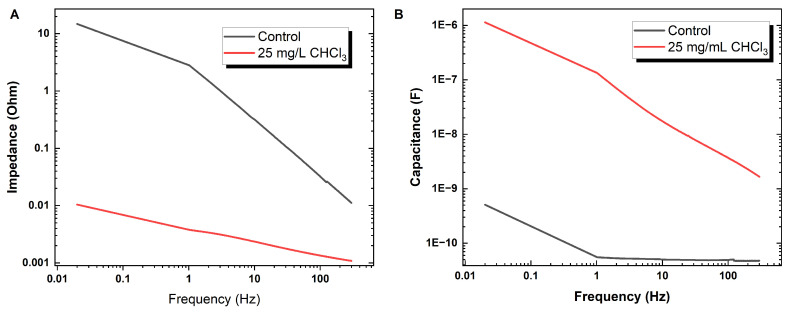
Frequency response of electrical characteristics in control and chloroform-exposed proteinoid samples. (**A**) Capacitance exhibits a logarithmic declining trend with increased frequency for both control (mean = 0.050 ± 0.027 nF) and 25 mg/L chloroform-exposed (0.087 ± 0.66 nF) cases. (**B**) Similarly, impedance progresses logarithmically for control (0.115 ± 0.875 Ω) and exposed (1.3956 × $10^{-3}$ ± 6.4324 × $10^{-4}$
Ω) conditions. In both metrics, chloroform introduction sharpens dynamic range and reduces the regularity of impedance spikes observed inherently, indicating regulated energy storage and transfer capacities. Recording dose-responsive trait alterations as frequency spectra shift could direct intentional tuning of electrical properties for specialised bioelectronic applications.

**Figure 10 biomimetics-09-00380-f010:**
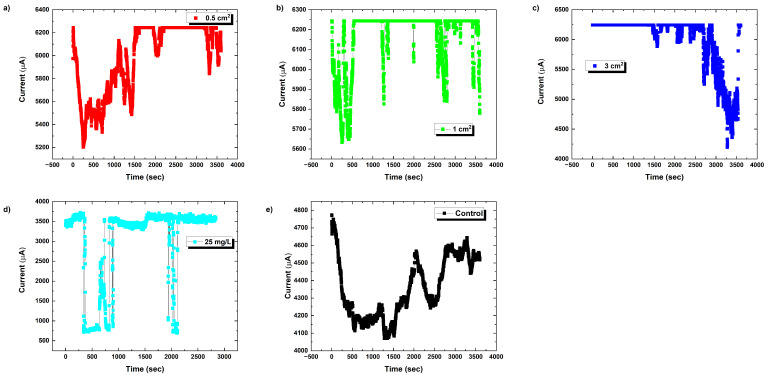
The chronoamperometric current responses of L–glutamic acid:L–arginine (L-Glu:L-Arg) proteinoids were measured over time as the chloroform (${\mathrm{CHCl}}_{3}$) concentrations increased. A control was included for comparison. The equilibrium duration is 0 s, the applied DC voltage is 0.1 volts, and the sampling interval is 1 s. Proteinoids exhibit temporary increases in conductivity that are influenced by changes in solvent levels, resulting from alterations in conformation and morphology. The rapid relaxation of memristive dynamics within subsecond timeframes suggests the stability of peptide microspheres. The graphs show the current response over time for proteinoid samples exposed to different chloroform presentations: (**a**) 0.5 cm^2^ filter paper embedded in chloroform; (**b**) 1 cm^2^ filter paper embedded in chloroform; (**c**) 3 cm^2^ filter paper embedded in chloroform; (**d**) 25 mg/L chloroform solution; (**e**) control (no chloroform exposure). The *x*-axis represents time (in seconds), and the *y*-axis shows the measured current (in amperes). Observed changes in current response indicate alterations in the electrochemical properties of proteinoids due to chloroform exposure.

**Figure 11 biomimetics-09-00380-f011:**
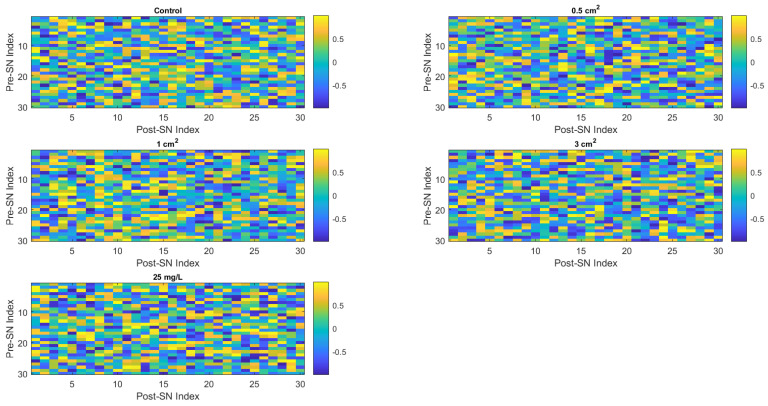
Heat maps of initial synaptic weights for various conditions. Each of the five subplots depicts the initial synaptic weights for each condition: control, 0.5 ${\mathrm{cm}}^{2}$, 1 ${\mathrm{cm}}^{2}$, 3 ${\mathrm{cm}}^{2}$, and 25 mg/L chloroform exposure. The synaptic weights are generated at random and displayed as a colour-coded matrix, with the colour intensity corresponding to the synaptic weight value. The *x* and *y* axes show postsynaptic and presynaptic neuron indices, respectively. These heat maps provide a preliminary overview of neural activity for every scenario.

**Figure 12 biomimetics-09-00380-f012:**
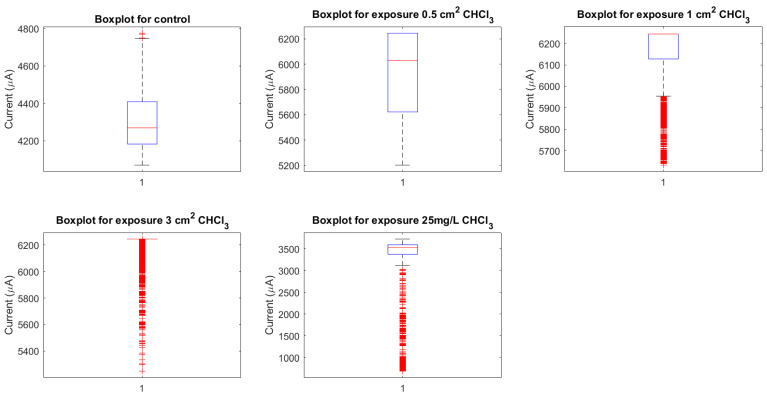
Current distributions under different conditions. The image depicts boxplots of the currents for five conditions: control, 0.5 ${\mathrm{cm}}^{2}$, 1 ${\mathrm{cm}}^{2}$, 3 ${\mathrm{cm}}^{2}$, and 25 mg/L chloroform exposure of proteinoids. Each box’s central line represents the median current, while the top and bottom bounds represent the upper (Q3) and lower (Q1) quartiles, respectively. The whiskers extend to the most extreme data points that are not deemed outliers, which are represented individually as dots. The notches around the median represent the median’s uncertainty. Mean currents (A) for control, 0.5 ${\mathrm{cm}}^{2}$, 1 ${\mathrm{cm}}^{2}$, 3 ${\mathrm{cm}}^{2}$, and 25 mg/L are around 4305, 5931, 6157, 6202, and 3046 μA, respectively, indicating a considerable drop in the current following chloroform exposure at 25 mg/L.

**Figure 13 biomimetics-09-00380-f013:**
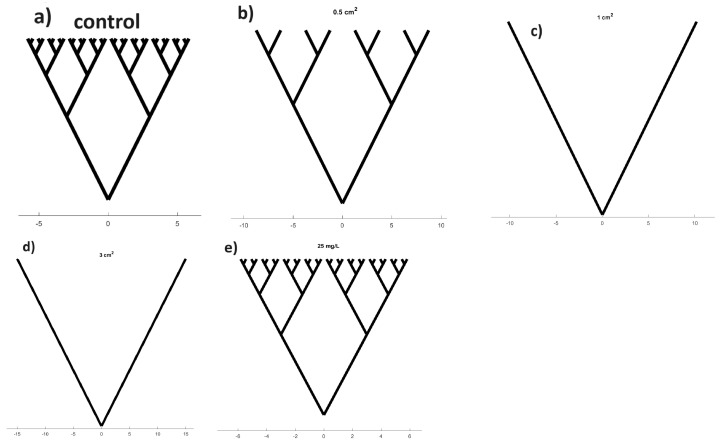
Binary tree pathways conceptually describe changes in the complexity of the interpreted proteinoid network following chloroform exposure. The subfigures depict the following conditions: (**a**) control, (**b**) 0.5 ${\mathrm{cm}}^{2}$, (**c**) 1 ${\mathrm{cm}}^{2}$, (**d**) 3 ${\mathrm{cm}}^{2}$, and (**e**) 25 mg/L. As the concentration of chloroform increases, trees undergo a transition from initially having more complex branching patterns (**a**) to simpler forms with fewer forks (**b**,**d**). This suggests that there is an emergence of enhanced structural order in proteinoid assemblies. However, when exposed to a concentration of 25 mg/L, the branches undergo recomplexification, indicating a possible degradation of organised dynamics under extreme conditions. Reduced maze-like trees, albeit abstract, symbolise the observed increase in order resulting from solvent interactions, which could perhaps be attributed to the molecular initiation of self-construction. The variations in complexity observed at different levels of exposure emphasise the ability to adjust proteinoid computational structures through external chemical regulation.

**Figure 14 biomimetics-09-00380-f014:**
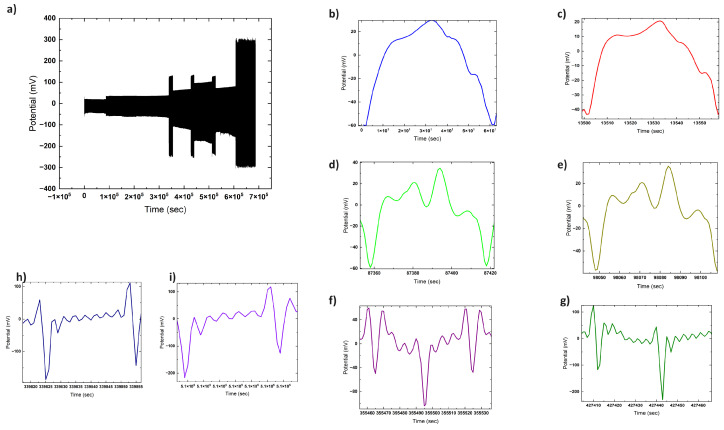
Sample recordings of output voltage signals generated by a 25 mg/L chloroform-modulated proteinoid assembly acting as a voltage-stimulated proto-brain system. Trace (**a**) shows the overlay of complex output signal over time containing multiple embedded frequency components. Enlargements highlight specific output signal harmonics parsed from the composite response: (**b**) first (fundamental) harmonic, (**c**) second harmonic, (**d**) fourth harmonic, (**e**) fifth harmonic, (**f**) eighth harmonic, (**g**) fourteenth harmonic, (**h**) sixteenth harmonic, and (**i**) thirteenth harmonic. Presence of higher-order mapped harmonics in the proteinoid voltage output demonstrates nonlinear transformational responses analogous to neural system kernels. Sensitive input tuning may enable selective filtering and computational pathway gating by harmonically complex bioderived signal-processing elements.

**Figure 15 biomimetics-09-00380-f015:**

Anesthesia of proteinoids using chloroform.

**Table 1 biomimetics-09-00380-t001:** Energy-minimisation results for the interaction between chloroform and the dipeptide L-Glu-L-Arg. The minimisation terminated normally at iteration 2511 because the gradient norm is less than the minimum gradient norm. The table presents the contributions of various energy components to the total energy of the system at the minimum configuration.

Energy Component	Energy (kcal/mol)
Stretch	0.6971
Bend	5.7351
Stretch–bend	0.2346
Torsion	−0.5590
Non-1,4 VDW	−9.7724
1,4 VDW	7.9509
Dipole/Dipole	−14.7107
Total energy	−10.4245

**Table 2 biomimetics-09-00380-t002:** Spike potential metrics across conditions.

Condition	Mean Pot (mV)	Std Dev	Max	Min	Kurtosis	Skewness
Control	0.895	0.419	1.909	0.361	3.91	1.37
0.5 cm^2^	0.281	0.059	0.412	0.209	2.83	0.87
1 cm^2^	0.104	0.037	0.173	0.047	2.26	0.31
3 cm^2^	0.085	0.047	0.274	0.049	14.97	3.58
25 mg/mL	0.065	0.028	0.134	0.027	5.05	1.39

**Table 3 biomimetics-09-00380-t003:** Interspike period metrics across conditions.

Condition	Mean Period (s)	Std Dev	Max	Min	Kurtosis	Skewness
Control	1392.86	425.83	1952.38	595.24	1.95	−0.36
0.5 cm^2^	1344.85	321.53	1871	882	1.82	0.41
1 cm^2^	923	356.92	1360	230	2.67	−0.87
3 cm^2^	1008.75	240.08	1505	718	3.46	0.95
25 mg/mL	228.2	223.91	623	69.8	2.08	0.92

**Table 4 biomimetics-09-00380-t004:** Lempel–Ziv complexity under different chloroform exposure conditions.

Condition	LZ Complexity
Control	4.739922
0.5 cm^2^	2.790569
1 cm^2^	1.383151
3 cm^2^	1.002987
25 mg/L	4.578150

**Table 5 biomimetics-09-00380-t005:** Comparative analysis of computational properties of proteinoids with and without anaesthetic retardation.

Property	Proteinoids	Anesthetised Proteinoids
Electrical Activity	Spontaneous spikes and oscillations	Suspended spiking, muted signals
Network Connectivity	Variable interproteinoid communication	Disrupted connections
Information Processing	Complex parallel logic and learning	Restricted processing capability
Handling	Requires insulation and interference avoidance	Permits direct manipulation and patterning
Reversibility	Inherent metastable state	Effects reversed upon chloroform removal

## Data Availability

The datasets generated and analysed during the current study are available in the zenodo repository, which can be accessed through the following web address: [https://zenodo.org/records/10443058 (accessed on 29 December 2023)].

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
