# Peer review of "On Effect of Chloroform on Electrical Activity of Proteinoids"

_biomimetics, 2024, doi:10.3390/biomimetics9070380_

Round 1
Reviewer 1 Report
Comments and Suggestions for Authors
It is interesting to develop simple molecular systems to artifically mimic complex biological activities. Basically, I think it could be accepted after considering the following points.
1. If possible, the authors give a graphical abstract including chemical strucutures (no cartoon) to demonstrate the desgin and work mechanism of this biomimetic system.
2. The detailed characterizations of interactions between dipeptide and chloroform at molecular level are missing. It is very of vital importance to versify the dynamic behavior of the system.
3. There are many errors in the maintext. So, please revise them carefully.
4. Figure 4 (SEM images) and Figure 12 are of low quality. It is hard to see them clearly. In addition, relevant TEM images are required to give the obvious change.
5. It will be much better for the authors to reorganize Figure 19-27 and Figure 5-9 in a concise manner. It is difficult to read this version.
6. In view of Chemistry, the approach can be extended to other dipeptide-based systems?
Comments on the Quality of English LanguageEnglish is good
Author Response
Dear Reviewer 1,
I wanted to inform you that I have uploaded the response to reviewers letter in relation to your document. I appreciate the time and effort you took to review the document.
Warm regards
Dr Mougkogiannis

Reviewer 2 Report
Comments and Suggestions for Authors
1. The meaning of Fig. 2 is unclear.
2. Experimental procedure should be detailed using some illustrations or photos of experimental set up.
3. The amplitude of potential changes observed in this work is extremely small. I've investigated the electrolytic solution potential change for the past two decades and observed such a small potential amplitude so many times. This potential change may be merely a potential fluctuation inevitably induced in this kind of experimental work. I never distrust the experimental results shown in this paper, but I cannot say that the interpretations for the potential profiles shown in this paper is genuinely right.
I don't want to reject this work, since this small potential amplitude may truly have some fundamentally important scientific meaning. But I cannot say that the authors' interpretations for this potential behavior is right, either.
I need more convincing data.
Author Response
Dear Reviewer 2,
I wanted to inform you that I have uploaded the response to reviewers letter in relation to your document. I appreciate the time and effort you took to review the document.
Warm regards
Dr Mougkogiannis

Round 2
Reviewer 2 Report
Comments and Suggestions for Authors
Still I cannot well figure out the experimental setup structure. Maybe the authors used a setup which is quite common for their research field. But I cannot figure it out.
How is the proteinoid and electrodes are arranged in the experimental system?
How is the heating block placed to the proteinoids?
In section 3.4, “Fig. ??” appears. It should be fixed.
Sorry, still I cannot figure out how the authors perform this experiment. I can understand that the authors emphasis that chloroform influences on the potential characteristics of microsphere.
Spike generation per se is intriguing, but it is not a new finding, since the spike generation of microsphere was discovered decades ago.
As I already questioned in the 1st round review process, the signals the authors study is quite low from my view. They look almost negligible signals. The authors’ discussion is not so convincing from my view.
This work may be beyond my knowledge. Therefore, I’m not willing to give a decision of Reject but I don’t have strong reason to give the decision of Accept.
Author Response
You can find attached the response to reviewer.
